# Interplay of Ferroptosis and Cuproptosis in Cancer: Dissecting Metal-Driven Mechanisms for Therapeutic Potentials

**DOI:** 10.3390/cancers16030512

**Published:** 2024-01-24

**Authors:** Jinjiang Wang, Jiaxi Li, Jiao Liu, Kit-Ying Chan, Ho-Sze Lee, Kenneth Nansheng Lin, Chi-Chiu Wang, Tat-San Lau

**Affiliations:** 1Department of Obstetrics & Gynaecology, The Chinese University of Hong Kong, Shatin, Hong Kong; wangjinjiang@link.cuhk.edu.hk (J.W.); kennethlinns@gmail.com (K.N.L.); ccwang@cuhk.edu.hk (C.-C.W.); 2Department of Surgery, Li Ka Shing Faculty of Medicine, The University of Hong Kong, Pokfulam, Hong Kong

**Keywords:** cancer, ferroptosis, cuproptosis, mitochondria, clinical trials, novel treatment

## Abstract

**Simple Summary:**

In the complex world of cancer, iron and copper play essential roles as trace metal ions that are crucial for cancer cell survival. Disruption in their metabolic functions can be lethal to cancer cells, triggering ferroptosis and cuproptosis, respectively. Given an accelerated proliferation rate, cancer cells exhibit a heightened dependence on iron and copper, exposing vulnerabilities that could potentially be exploited to reverse drug resistance. Notably, mitochondria, the cellular powerhouses, play a crucial role in regulating both ferroptosis and cuproptosis. This review focuses on elucidating the key mechanisms behind ferroptosis and cuproptosis and summarizes recent clinical applications targeting dysfunctional iron and copper metabolic pathways. Drug resistance is a hallmark of cancer development that underscores a critical need to address it, underscoring the critical need to explore novel approaches. Understanding and targeting these metal-related processes offers promising approaches for developing innovative cancer therapies, making use of vulnerabilities specific to cancer cells.

**Abstract:**

Iron (Fe) and copper (Cu), essential transition metals, play pivotal roles in various cellular processes critical to cancer biology, including cell proliferation, mitochondrial respiration, distant metastases, and oxidative stress. The emergence of ferroptosis and cuproptosis as distinct forms of non-apoptotic cell death has heightened their significance, particularly in connection with these metal ions. While initially studied separately, recent evidence underscores the interdependence of ferroptosis and cuproptosis. Studies reveal a link between mitochondrial copper accumulation and ferroptosis induction. This interconnected relationship presents a promising strategy, especially for addressing refractory cancers marked by drug tolerance. Harnessing the toxicity of iron and copper in clinical settings becomes crucial. Simultaneous targeting of ferroptosis and cuproptosis, exemplified by the combination of sorafenib and elesclomol-Cu, represents an intriguing approach. Strategies targeting mitochondria further enhance the precision of these approaches, providing hope for improving treatment outcomes of drug-resistant cancers. Moreover, the combination of iron chelators and copper-lowering agents with established therapeutic modalities exhibits a synergy that holds promise for the augmentation of anti-tumor efficacy in various malignancies. This review elaborates on the complex interplay between ferroptosis and cuproptosis, including their underlying mechanisms, and explores their potential as druggable targets in both cancer research and clinical settings.

## 1. Introduction

Iron and copper play crucial roles in tumor proliferation and metastasis, but their disrupted uptake and distribution can induce toxicity in cancer cells [1,2]. Perturbations in iron and copper homeostasis precipitate the occurrence of ferroptosis, defined by the iron-dependent accumulation of lipid peroxides, proposed in 2012, and cuproptosis, characterized by the copper-triggered modality of mitochondrial cell death, introduced in 2022 [3,4,5]. These mechanisms play specific roles, with ferroptosis influencing tumor growth in breast cancer [6], liver cancer [7], and ovarian cancer [8], and cuproptosis mechanisms being scrutinized in colorectal cancer [9] and lung cancer [10]. This intricate investigation into the functional roles of iron and copper contributes to a better understanding of their intricate influences across a spectrum of diseases, including cancers [11,12,13]. By delving deeply into the interconnection between ferroptosis and cuproptosis, this investigation offers unique scientific perspectives, shedding light on the complex molecular mechanism that underlies these metal-associated cellular processes in cancers. Furthermore, a comprehensive exploration delves into the exploitation of mechanisms of ferroptosis and cuproptosis, with a specific focus on the roles played by iron and copper. This unveils novel perspectives, extending beyond conventional viewpoints and delving into the dynamic interplay of these trace elements in cancer biology. The in-depth exploration of iron and copper intricacies holds the promise of uncovering novel insights. This review not only summarizes potential therapeutic targets but also highlights the remarkable adaptability and resilience exhibited by cancer cells within their microenvironment [14,15]. The modulation of ferroptosis and cuproptosis emerges not merely as a revolution in therapeutic approaches but also as a pioneering frontier in precision medicine [16,17], ushering in an era of tailored interventions that could substantially elevate the precision and efficacy of anticancer treatments [18,19]. This signifies a notable advance in personalized oncology, directing focus towards unraveling cancer’s complex molecular landscape through interventions focusing on iron and copper mechanisms in clinical settings [17,20]. Nonetheless, the utilization of iron and copper-induced cell death in cancer therapy introduces adverse effects on normal cellular function [21,22] and perturbs the equilibrium of adjacent T cells and macrophages, thereby bringing out the intricate interplay between metal-induced cytotoxicity and immune responses [23,24]. Through integrating molecular insights and detailed case studies and summarizing current knowledge with future perspectives, we elucidate the significance of iron and copper in cancer. Subsequently, we explore the distinct mechanisms underlying ferroptosis and cuproptosis, providing updated insights into the intermediary role of mitochondria in connecting these pathways. Moreover, we present a perspective on cancer therapies designed to overcome drug resistance and optimize existing treatments by selectively targeting ferroptosis and cuproptosis.

## 2. Iron/Copper in Cancers

Iron dysregulation in malignancy suggests therapeutic potential via depletion and ferroptosis induction [25]. Currently, in clinical investigations, these strategies may be combined with existing treatments for metastatic cases. In contrast, copper undergoes precise regulation to prevent oxidative stress and cellular dysfunction. Ongoing research has revealed the potential therapeutic strategies for inducing copper imbalance [26].

### 2.1. Iron Dysregulation in Cancer Cells

The dysregulation of iron metabolism, characterized by cancer cells’ heightened dependency, termed “iron addiction”, can increase the oncogenic risk and promote tumor growth, with recent research uncovering the cellular processes, including ferroptosis, controlled iron efflux, and ferritinophagy. The alterations in iron metabolism observed in both high-grade serous ovarian cancer (HGSOC) and HGSOC tumor-initiating cells present potential targets for ovarian cancer treatment [27]. Essential for cellular functions, iron in excess can lead to diseases. Dysregulated iron, associated with conditions like infections and cancer, promotes the development of purpose-designed chelators and hepcidin agonists as potential therapies, with a focus on cancer cells through iron-driven reactive oxygen species generation with ferroptosis, providing a novel therapeutic avenue [28]. Elevated iron levels are detected in ovarian cancer [29], breast cancer [30], and liver cancer [31]. Patients with hereditary hemochromatosis have a 20–200 times higher risk of liver cancer than that of the general population. Even slight hepatic iron accumulation significantly contributes to hepatocellular carcinoma (HCC) development in chronic hepatitis C. Phlebotomy, reducing serum ferritin and hepatic 8-hydroxy-2′-deoxyguanosine content, lowers the risk of HCC progression. Cancer cells, often resistant to therapy due to impaired cell death mechanisms, exhibit heightened dependence on iron, which is crucial for their growth. Compounds including ML162, withaferin A, and the FDA-approved anticancer agent altretamine have demonstrated the capacity to induce ferroptosis through the mechanism of glutathione peroxidase 4 (GPX4) inactivation [32].

### 2.2. Copper Dysregulation in Cancer Cells

Cancer cells exhibit a high dependency on copper, akin to their iron addiction. The clinically-approved drug metformin selectively eradicates persister cancer cells by targeting mitochondrial copper, inducing reactive oxygen species, mitochondrial dysfunction, and apoptosis, while also blocking the epithelial-to-mesenchymal transition, a process that is crucial for persister cancer cell genesis [33]. Elevated copper levels in cancer cells play a crucial role in controlling tumor growth, suggesting that manipulating copper metabolism could be a promising strategy for tumor treatment. The copper chelator tetrathiomolybdate (TTM) treatment hinders breast cancer metastasis by lowering Cu2+ levels and suppressing Mediator Of Cell Motility 1 (MEMO1) expression and angiogenesis [34]. Copper’s dual role as an essential component for cancer growth and metastasis, influencing signaling pathways and cell death mechanisms, provides novel options for targeted prostate cancer therapies. TTM efficiently inhibits the growth of androgen receptor (AR)-expressing lymph node carcinoma of the prostate cells [35,36]. As a rate-limiting nutrient for cancer cell growth, copper has demonstrated anti-neoplastic effects through oral copper chelators, revealing potential targets for anticancer therapies related to pathways of oncogenic signaling and drug resistance [37].

### 2.3. Iron and Copper Labile Pool

Nuclear factor erythroid 2-related factor 2 (NRF2) inhibition stands out as a potential strategy to elevate the labile iron pool (LIP), intensifying ferroptotic sensitivity in cancer cells that is clinically relevant to ovarian cancer [38]. PTEN-induced kinase 1 (PINK1) orchestrates mitophagy and ferritinophagy, ensuring intracellular iron homeostasis crucial for the survival and growth of colorectal cancer cells [39]. As a transient and redox-active iron reservoir, LIP is assessed using novel fluorescent metalosensors, offering insights into its dynamic levels and roles in specific subcellular compartments beyond traditional iron regulatory pathways. Newly developed reactivity-based probes enable the precise and sensitive detection of labile iron pools in tumors, offering insights into iron metabolism and disease-related alterations [40]. Labile iron significantly promotes HepG2.2.15 cell growth via L-type calcium channels, suggesting novel approaches for liver cancer detection and alternative therapeutic strategies like calcium channel blockers and specific iron chelators [41]. On the other hand, limiting the labile copper pool emerges as a potential strategy to inhibit cancer growth by restraining MAP kinase kinase1/2 (MEK1/2) activity, even in the presence of activating mutations. This demonstrated copper modulation through chelators or ionophores as a promising avenue for regulating MAPK signaling in cancer, especially in combinational therapies [42].

## 3. Ferroptosis and Cuproptosis

### 3.1. Ferroptosis

Ferroptosis, a distinctive iron-dependent form of programmed cell death, was discovered by Brent R. Stockwell’s lab in 2012, with GPX4 as the core regulator of ferroptosis [43]. Triggered by the perturbation of pathways related to iron metabolism, lipid peroxidation, and antioxidant defense, ferroptosis is involved in tumor development and responses to anticancer therapies. Agents like erastin, RSL3, approved drugs, ionizing radiation, and cytokines induce ferroptosis, suppressing tumor growth; however, they may induce inflammation-associated immunosuppression, fostering tumor development [44]. The impact of ferroptosis on tumor biology, influenced by mutations in cancer-related genes and stress response pathways, remains incompletely understood but holds potential in systemic therapy, radiotherapy, and immunotherapy.

#### 3.1.1. Iron Metabolism

Iron metabolism, which is intricately regulated, plays a crucial role in ferroptosis. Disruption in iron metabolism leading to an elevated iron level offers potential therapeutic avenues in cancer treatment, such as iron chelation [45]. The labile iron pool, a dynamic cellular reservoir, crucially influences the cellular processes of iron metabolism by modulating reactive oxygen species (ROS) levels and redox status. Proteins like transferrin, ferritins, hepcidin, and ferroportin (FPN) are essential for systemic iron homeostasis [46]. Notably, the hepcidin–FPN axis emerges as the principal regulator governing extracellular iron balance [47].

Altered iron homeostasis and excessive production of reactive oxygen or nitrogen species can directly induce ferroptosis by catalyzing the oxidation of phospholipids in cell membranes. The fluorescent iron probe-1 (FIP-1) is capable of detecting changes in the LIP in live HEK 293T cells during iron chelator treatment, enabling the observation of alterations in labile iron levels in cells undergoing ferroptosis [40,48,49]. Nanotherapeutics, as highlighted in studies such as “Targeting iron metabolism in cancer therapy”, demonstrate the potential of nanomedicine to regulate iron metabolism in cancer biology, revealing the intersection of iron metabolism with ferroptosis in cancer pathogenesis and treatment [50,51].

#### 3.1.2. Lipid Peroxidation and ROS

Lipid peroxidation can be initiated through either non-enzymatic or enzymatic processes. Non-enzymatic lipid peroxidation is fueled by radicals, impacting various lipid constituents such as glycolipids, phospholipids (PLs), and cholesterol. Enzymes like lipoxygenases (LOXs) and cyclooxygenases (COXs) can also catalyze the oxidation of lipids. Importantly, LOX activation is intricately correlated with the cellular redox state, including its association with glutathione (GSH) levels [52].

ROS-induced lipid peroxidation is a crucial factor in ferroptosis, a conserved mechanism that has an excess of ROS triggering biomembrane damage, propagating lipid peroxidation chain reactions, and determining various cell death outcomes based on the integration of pro-survival and pro-death signals from subcellular organelles [53,54]. The imbalance of these lipid hydroperoxides leads to cellular membrane damage and ferroptosis. While polyunsaturated fatty acids (PUFAs) are predominantly involved in membrane phospholipid oxidation, saturated and monounsaturated fatty acids (SFAs/MUFAs) also play a role in lipid peroxidation and ferroptosis [55,56]. Key enzymes, Acyl-CoA Synthetase Long Chain Family Member 4 (ACSL4), and Lysophosphatidylcholine Acyltransferase 3 (LPCAT3), drive lipid peroxidation by enriching membranes with vulnerable fatty acids, making them susceptible to oxidative damage. The balance among the incorporation, oxidation, and neutralization of PUFA is critical for cellular sensitivity to ferroptosis [57,58], highlighting therapeutic potential in cancer treatment.

#### 3.1.3. Key Molecules Governing Ferroptosis

GPX4, a key enzyme in ferroptosis [59,60], plays a crucial role in averting iron-dependent toxic lipid ROS formation by converting lipid hydroperoxides to lipid alcohols. The inhibition of GPX4 function induces lipid peroxidation, eventually leading to ferroptosis [61]. The intracellular redox imbalance triggers ROS production, with the cystine/glutamate antiporter (System Xc-), GPX4, and GSH forming the critical System Xc-/GSH/GPX4 axis to prevent lipid peroxidation-mediated ferroptosis. The inhibition of this critical axis emerges as a promising strategy for treating tumors, especially those that are drug-resistant, which offers a novel therapeutic choice in combination with chemotherapeutic agents [62,63]. The eminent tumor suppressor gene p53 executes its function in ferroptosis by modulating GSH and GPX4 [64,65] (Figure 1). Chemotherapy-resistant cancer cells with heightened iron dependency exhibit vulnerability to ferroptosis upon GPX4 inhibition, providing a promising therapy for effective cancer treatment [16]. The repression of SLC7A11 renders HCC cells susceptible to GPX4 inhibition, while inhibitors of system Xc- or GPX4 selectively eliminate cancer cells but preserve healthy ones. TGF-β1 enhances susceptibility to GPX4 inhibition in well-differentiated HCC cells, suggesting a potential therapeutic strategy for selective ferroptosis induction in specific tumor sub-populations [66,67]. This approach holds the potential for more effective and less toxic cancer treatments by targeting cancer cell metabolism. Among these, ferroptosis is tightly regulated by central cellular signaling pathways. The Nrf2-ARE pathway, governed by Keap1, serves as a key defense mechanism, with Nrf2 acting as a negative regulator of ferroptosis [68]. FSP1, a target of Nrf2, independently inhibits ferroptosis by reducing ubiquinone [69,70]. The Hippo pathway, along with YAP/TAZ activation, emerges as a determinant of ferroptosis, especially in cancer cells [71]. Heat shock proteins, particularly HSPB1, play a role in regulating iron uptake and limiting lipid peroxidation, contributing to the complex regulatory network of ferroptosis [72].

### 3.2. Cuproptosis

Cuproptosis is triggered by opper overload [4]. As the copper ion balance is imperative for cellular health, any imbalance is likely to induce cell death or even contribute to carcinogenesis [26,73]. This section delves into the phenomenon of cuproptosis, emphasizing the accumulation of intracellular copper and its influence on copper dynamics, mitochondrial metabolism, lipoylated proteins in the tricarboxylic acid (TCA) cycle, iron-sulfur cluster proteins, and oxidative phosphorylation.

#### 3.2.1. Intracellular Copper Accumulation

Excess copper overload in cells can lead to cuproptosis, a process associated with ROS production and oxidative stress. Elesclomol is a mitochondrion-targeting agent designed for treating diseases, including cancers and copper-associated disorders [74]. Elesclomol binds to copper in the extracellular environment, leading to intracellular copper accumulation [75,76]. ATP7A and ATP7B, which are essential for copper homeostasis, regulate the function of cuproenzyme and mediate excess copper excretion, which is critical for preventing disorders like Menkes and Wilson diseases [77]. What is more, CTR1 (SLC31A1) codes for a high-affinity copper-uptake protein crucial for delivering copper to mammalian cells [78]. Notably, ATP7A is essential for cuproenzyme generation and plays a key role in copper export, particularly in enterocytes, affecting cancer drug resistance, prognosis, and therapeutic consequences [78,79] (Figure 1).

#### 3.2.2. Aggregation of Lipoylated Proteins and Destabilization of Fe-S Cluster Proteins

The aggregation of mitochondrial lipoylated proteins is a distinctive feature of cuproptosis, where copper-induced proteotoxic stress, particularly with ferredoxin 1 (FDX1) and Dihydrolipoamide S-acetyltransferase (DLAT) involved, disrupts the TCA, leading to mitochondrial dysfunction and the loss of iron–sulfur (Fe–S) cluster proteins [80]. Lipoylation, a preserved lysine posttranslational modification, is observed in only four essential multimeric metabolic enzymes. The aberration of these mitochondrial proteins is associated with cancers [81]. Fe–S clusters, integral for DNA metabolism and electron transport, become vulnerable in cancer through a hyperactive biogenesis pathway. Targeting this pathway demonstrates a strategy to specifically sensitize cancer cells while safeguarding normal cells due to their distinctive iron dependence [82]. The direct binding of copper to lipoylated TCA cycle proteins results in their disulfide-bond-dependent aggregation, resulting in the degradation of Fe–S cluster proteins, ultimately contributing to cuproptosis [26,83] (Figure 1).

#### 3.2.3. Oxidative Phosphorylation Disruption

Cuproptosis disrupts oxidative phosphorylation, leading to mitochondrial dysfunction and cell death [73]. Cells utilizing glycolysis for energy production exhibit resistance to cuproptosis, while those dependent on the TCA cycle and oxidative phosphorylation are sensitive [80]. This distinction is notable in cancer cells, which prefer glycolysis, contributing to their resistance to cuproptosis for cancer cells. Hypoxia-induced reliance on glycolysis further reduces the sensitivity of cancer cells to cuproptosis, which is distinct from ferroptosis, which requires glucose uptake and pyruvate oxidation [84]. This occurs mainly in cells relying on oxidative phosphorylation for energy. The aggregation of lipoylated proteins and reduced Fe-S clusters disrupt the mitochondrial electron transport chain (mtETC) and TCA cycle [85,86]. It has been noted that the inhibitors of oxidative phosphorylation might inhibit cuproptosis by suppressing the protein stress response [87].

## 4. The Role of Mitochondria in Ferroptosis and Cuproptosis

Ferroptosis contains mitochondrial changes, leading to altered energy metabolism and oxidative stress, while cuproptosis, triggered by intracellular copper overload, induces mitochondrial proteotoxic stress via FDX1, resulting in mitochondrial dysfunction. Targeting mitochondrial functions offers potential therapeutic interventions to modulate both ferroptosis and cuproptosis for diseases including cancer.

### 4.1. Mitochondria and Ferroptosis

The interplay between mitochondria and ferroptosis is evident, as they collaboratively govern iron metabolism and lipid peroxidation. This dynamic interaction prompts a metabolic rearrangement marked by a decrease in glycolysis and alterations in oxidative phosphorylation. This highlights the pivotal contribution of mitochondria as the primary ATP producers in orchestrating these processes in cancer [88]. Excessive stress causes irreversible damage to mitochondria, disrupting their structure and function. These changes include an increase in membrane density, damage to mitochondrial cristae, and a more condensed mitochondrial membrane density [89,90]. Additionally, in ferroptosis, mitochondria undergo a reduction in volume, showing smaller sizes in comparison to their normal state. This is coupled with a notable decrease and, in some cases, the complete disappearance of mitochondrial cristae and the outer membrane [91]. Crucial components of this modulation include processes such as mitochondrial fusion, fission, and mitophagy [92]. Specific mitochondrial lipids, such as cardiolipin, serving as direct substrates for lipid peroxidation, build a connection between mitochondrial vitality and ferroptosis. The peroxidation of cardiolipin results in the damage and permeabilization of the mitochondrial membrane, leading to structural and functional impairment [93].

### 4.2. Mitochondria and Cuproptosis

Cuproptosis begins with copper accumulation in the cytoplasm and organelles, triggering the aggregation of mitochondrial lipoylated modules and destabilizing Fe-S cluster proteins that are critical for maintaining mitochondrial function [73,80]. Cuproptosis, in contrast to ferroptosis, is characterized by its dependence on galactose-mediated mitochondrial respiration. This preference significantly affects cells relying on the TCA cycle more than those dependent on glucose-induced glycolysis, serving as a key distinguishing feature for the two cell death pathways [4,94]. Cuproptosis is intricately associated with mitochondrial respiration, with its sensitivity influenced by the state of mitochondrial respiratory activity [80,95]. There are two mitochondrial proteotoxic stress pathways involved in cuproptosis. Copper overload induces enhanced mitochondrial protein lipoylation, leading to the aggregation of lipoylated DLAT via disulfide bonds. The accumulation of toxic lipoylated DLAT, mediated by FDX1, coupled with the degradation of Fe-S cluster proteins [80], contributes to the mechanism of unique cellular demise, which is a typical characteristic of cuproptosis (Figure 1).

### 4.3. Mitochondrial Interplay in Ferroptosis and Cuproptosis

Mitochondria act as potential metal pools, and the mitochondrial respiratory chain works as an oxygen sensor under hypoxic conditions. Due to their distinct mechanisms related to mitochondrial metabolism, ferroptosis and cuproptosis show distinct vulnerabilities when exposed to hypoxia [80,96]. Sorafenib and erastin, known as ferroptosis inducers, enhance protein lipoylation by inhibiting FDX1 degradation via mitochondrial proteases. Simultaneously, they reduced intracellular copper chelator GSH synthesis by blocking cystine imports [11]. Iron drives lipid peroxidation, culminating in ferroptosis, whereas copper binds and disrupts DNA, activating E2D2-induced protein ubiquitination and degradation. Additionally, copper stimulates protein lipoylation and aggregation, promoting cuproptosis [11]. Mitochondria rely on iron, copper, and calcium for efficient ATP production. The mitochondrial electron transport chain and other mitochondrial functions are modulated by these metal ions, affecting both ferroptosis and cuproptosis [97]. This also suggests a novel anticancer therapeutic strategy to co-target cuproptosis and ferroptosis by acting on the metabolism of mitochondria.

## 5. Targeting Ferroptosis/Cuproptosis to Overcome Drug Resistance

Cancer drug-resistant cells and cancer stem cells (CSCs) possess unique characteristics that make them susceptible to specific cell death mechanisms, offering potential strategies to overcome drug resistance and achieve tumor eradication [98]. Among these mechanisms, ferroptosis has been shown to be promising in targeting drug-resistant cancer cells, providing a targeted therapeutic approach to induce cell death and overcome resistance [99]. Targeting CSCs and triggering ferroptosis in these cells is considered essential for achieving complete tumor eradication [100]. The current research focuses on deciphering the association between ferroptosis and drug resistance in tumors, aiming to improve treatment outcomes by targeting the resistance mechanisms [101,102].

### 5.1. Drug Resistance Related to Iron/Copper

Drug resistance in cancer is a complex feature that can be influenced by disruptions in both iron and copper metabolism. The disruption of iron metabolism commonly occurs in cancer cells and often leads to multidrug resistance [103]. Iron, which is crucial for cell proliferation, is elevated in rapidly dividing, drug-resistant cells. However, iron chelators effectively overcome drug resistance by disrupting the overall iron metabolism in these cells [103]. An imbalance in ferroptosis, characterized by aberrant iron metabolism and lipid peroxidation, is a major contributor to chemotherapy resistance and tumor recurrence. Elevated levels of hydrogen peroxide in cancer cells can trigger ferroptosis through interactions with iron ions, and modulating ferroptosis through targeting redox homeostasis is beneficial for overcoming drug resistance [54,104].

Furthermore, copper has been demonstrated to be involved in tumor promotion and the development of drug resistance. A copper chelate was shown to be effective against drug resistance in previous studies that demonstrated copper’s potential role in drug resistance by analyzing serum copper levels in mice with drug-sensitive and drug-resistant tumors, compared to healthy mice without tumors [105]. Moreover, approaches targeting nano-drug-based cell cuproptosis are being explored to enhance the efficiency of anticancer therapeutics [106]. Prediction models of different types of cuproptosis have been constructed to guide these approaches. Drug-resistant tumors often exhibit enhanced mitochondrial metabolism, which is associated with cuproptosis and cancer progression [17]. Targeting cuproptosis using copper ion ionophores in combination with small-molecule drugs demonstrates a novel therapeutic strategy to overcome cancer drug resistance [83].

Understanding the interplay between iron and copper metabolism and their role in cancer cell drug resistance is crucial for effective anticancer therapies. Manipulating these pathways holds promise for conquering drug resistance and improving anti-tumor efficacy for cancer patients.

### 5.2. Persister Cells and CSCs

Persister cells and CSCs both play significant roles in drug resistance and tumor recurrence. Persister cells, as a subpopulation of cancer cells, can survive despite anticancer treatments, contributing to the development of resistance to certain therapies [107]. Similarly, CSCs are responsible for acquired resistance to conventional therapies and are involved in tumor heterogeneity [108]. CSCs exhibit inherent resistance to standard treatments and show phenotypic plasticity within tumors, rendering them challenging targets for conventional therapies [109]. Aldehyde dehydrogenase (ALDH)+ cancer stem cell populations hold strong self-renewal and tumor-initiating capabilities that allow cancer cells to develop resistance to chemotherapy and radiotherapy [110].

Both persister cells and CSCs exhibit unique metabolic states compared to bulk tumor cells. Persister cells have been found to have a higher dependency on iron and copper, making them vulnerable to ferroptosis and cuproptosis [111]. Targeting iron metabolism could induce ferroptosis in persister cells, offering a strategy to eradicate drug-resistant cancer cells. On the other hand, CSCs may be more susceptible to lipid peroxidation and ferroptosis due to potential alterations in lipid metabolism. Targeting the sensitivity of CSCs to ferroptosis could be a promising approach to enhance efficacy of cancer therapeutics.

Overall, both persister cells and CSCs represent sub-populations of cancer cells that contribute to drug resistance and tumor recurrence. Understanding the metabolic vulnerabilities of these cells to certain mechanisms, particularly their dependence on iron and copper metabolism, can provide valuable insights for the development of targeted therapies to overcome the drug resistance of cancer cells and improve treatment outcomes in cancers.

## 6. Clinical Trials Targeting Ferroptosis and Cuproptosis

In a clinical trial examining iron-overload cases with obesity and colorectal cancer (CRC), in spite of being a crucial nutrient, excess iron is associated with carcinogenesis, and CRC uniquely maintains access to two iron acquisition routes [112]. In a separate trial, the efficacy of intravenous ferric carboxymaltose and oral iron substitution was assessed in patients with metastatic colorectal cancer (mCRC) and iron deficiency anemia. Iron, whether in excess or insufficiency, may contribute to colorectal cancer pathogenesis, with an inverse association between serum iron level and the risk of colon cancers [113]. At present, there is a lack of extensive investigations regarding how iron metabolism is altered and how it influences the risk of CRC in the context of obesity from the perspective of a cancer expert (ClinicalTrials.gov, accessed on 1 December 2023) (Table 1).

Clinical trials focusing on cuproptosis and copper metabolism are less studied compared to those concentrating on iron metabolism, but ongoing research, particularly concerning diseases associated with copper imbalance, is actively investigating this subject (Table 2). In the ongoing phase II trial of disulfiram with copper in metastatic breast cancer, there is an endeavor to establish clinical evidence for the potential efficacy of disulfiram and copper as a therapeutic approach in metastatic breast cancer, particularly after the failure of conventional systemic and/or locoregional treatments (ClinicalTrials.gov, accessed on 1 December 2023). Disulfiram, recognized for alcohol aversion, exhibits anticancer effects and reduces cancer-related mortality in subjects with unremitting doses. The active metabolite, the ditiocarb–copper complex, selectively accumulates in tumors, and nuclear protein localization protein 4 (NPL4) was identified as the molecular target for its tumor-suppressing effects [114]. Disulfiram/copper (DSF/Cu) holds promise as an effective antitumor agent in clinical settings, owing to its impressive anticancer efficacy and safety [115]. Despite its recognized potential, the specific mechanism underlying the anticancer effects of DSF/Cu, particularly in copper metabolism in cancers, remains largely unknown.

## 7. Ferroptosis and Cuproptosis in Conventional Treatment 

In clinical oncology, the emerging strategy of integrating ferroptosis and cuproptosis alongside traditional cancer treatments holds great potential for dealing with cancer. This innovative approach aims to augment the efficacy of cancer therapy by capitalizing on the distinctive vulnerabilities that are inherent to cancer cells. Furthermore, the exploration of repurposing existing drugs to uncover their roles in iron and copper metabolisms stands at the forefront of pioneering strategies, offering new treatments for addressing refractory cancers.

### 7.1. Ferroptosis as a Novel Approach

Ferroptosis has emerged as a promising intervention option in clinical oncology [116,117]. Recent advances in understanding the molecular mechanisms of ferroptosis have paved the way for its exploration in various cancer settings. Notably, in glioblastoma multiforme (GBM), a highly aggressive brain tumor, ferroptosis inducers such as erastin and sulfasalazine have shown efficacy in preclinical studies. Neutrophil-triggered ferroptosis is implicated in GBM necrosis, revealing a pro-tumorigenic role of ferroptosis. These compounds target the vulnerabilities of GBM cells, triggering lipid peroxidation and subsequent cell death [118,119]. Furthermore, in pancreatic ductal adenocarcinoma (PDAC), a cancer notoriously resistant to conventional therapies, ferroptosis has been identified as a potential Achilles’ heel [120]. Glutaminase inhibitors, such as CB-839, have exhibited the ability to induce ferroptosis in PDAC cells by disrupting glutamine metabolism. Drugs targeting ferroptosis for PDAC treatment, like artesunate and zalcitabine, emphasize the need for modulating drugs to address resistance. In the context of oncogenic Kras mutations driving PDAC progression, recent findings reveal Slc7a11 deletion-induced ferroptosis hinders Kras-driven growth, wherein ferroptotic damage may enhance Kras-driven PDAC through the TMEM173-dependent DNA-sensing pathway in mice [121].

Tumor suppressor genes such as TP53 and KEAP1, which regulate cellular responses to oxidative stress, play pivotal roles in modulating ferroptosis [65]. Subsequently, ferroptosis suppressor protein 1 (FSP1) plays a crucial role in ferroptosis induced by cysteine depletion in KEAP1 mutant non-small-cell lung cancer (NSCLC) [122]. Decoding these genetic determinants allows for tailoring ferroptosis-based therapies to individual patient profiles. Other than solid tumors, ferroptosis modulation has shown promise in hematological malignancies as well [123]. For instance, acute myeloid leukemia (AML) cells exhibit sensitivity to ferroptosis inducers like RSL3. Erastin, RSL3, and ML162 have demonstrated the ability to induce ferroptosis in KBM7 cells [124]. Ferroptosis, despite its therapeutic potential, raises concerns due to its capability to cause tissue damage and inflammation [125,126]. A delicate balance is crucial for effective and safe clinical applications, requiring a nuanced understanding of precise regulation and the potential side effects [127]. Harnessing ferroptosis in clinical oncology holds great promise, offering novel strategies to overcome treatment resistance and improve therapeutic outcomes across various cancer types. 

Across diverse cancers, sorafenib demonstrates the ability to induce ferroptosis by promoting lipid peroxidation. In HCC, concurrent administration with ferroptosis inhibitors, such as Liproxstatin-1, aims at amplifying or modulating this response while mitigating potential toxicity. In HCC cells, induction of ferroptosis by sorafenib includes the ERK pathway, which facilitates the TRIM54-mediated FSP1 ubiquitination [128]. Regarding breast cancer, doxorubicin-induced cardiotoxicity may involve iron-dependent processes and oxidative stress, prompting co-administration with iron chelators to adjust iron levels and potentially induce ferroptosis [129]. In gastrointestinal stromal tumors, Imatinib has been found to induce ferroptosis through STUB1-mediated GPX4 ubiquitination [130]. In a prior study, we observed that cetuximab, an approved treatment for metastatic CRC with KRAS, enhanced RSL3-induced ferroptosis by suppressing the NRF2/HO-1 pathway in KRAS-mutant CRC [131]. The natural product β-elemene has been identified as a novel ferroptosis inducer, and a combination of β-elemene and cetuximab inhibited KRAS-mutant CRC cell growth by inducing ferroptosis [132]. An investigation of arsenic trioxide’s potential to induce ferroptosis in acute promyelocytic leukemia cells, likely through its impact on iron metabolism, explores co-administration with iron chelators to modulate this ferroptotic response [133]. The examples highlight the promising potential of inducing ferroptosis with various drugs across cancer types, paving the way for developing refined and targeted therapeutic strategies in cancer treatment.

### 7.2. Cuproptosis as a Novel Approach

In HCC, copper dysregulation has been identified as a crucial factor [134]. Cuproptosis can be induced in primary liver cancer cells by ferroptosis inducers like sorafenib and erastin. These agents enhance the aggregation of lipoylated proteins dependent on copper. The mechanism involves the upregulation of protein lipoylation by suppressing mitochondrial matrix-related proteases, particularly FDX1 protein degradation. Simultaneously, they reduce intracellular copper chelator GSH synthesis by inhibiting cystine import. This dual role reveals the significance of copper dynamics and cuproptosis in liver cancer [11]. TM, an oral copper chelator, demonstrates efficacy in high-risk triple-negative breast cancer (TNBC) by disrupting Complex IV and involvement of the AMPK/mTORC1 pathway [135]. This finding suggests a potential copper-metabolism-metastasis axis, highlighting TM’s therapeutic potential for high-risk TNBC. In melanoma, the copper ionophore elesclomol exhibits notable specificity against GNAQ/11-mutant uveal melanoma cells, retaining therapeutic efficacy in patient-derived tumor cells in an ex vivo model [135]. These examples underscoring cuproptosis in aspects of copper homeostasis, mitochondrial function, and other cell death pathways provide a multifaceted perspective for therapeutic interventions. In the clinical landscape, exploring drugs that selectively modulate copper levels or disrupt specific steps of the cuproptotic pathway holds promise for enhancing anticancer treatment responses.

## 8. Conclusions and Perspectives

This review thoroughly explores the intricate interplay among iron and copper and the regulatory mechanisms governing ferroptosis and cuproptosis in cancers. In ferroptosis, the emphasis on iron-dependent lipid peroxidation provides a promising target, with potential applications in refining personalized interventions tailored to the specific iron metabolism of individual cancers. Conversely, cuproptosis, driven by copper overload, presents a unique approach to cancer therapeutics, highlighting copper’s role in disrupting mitochondrial functions and inducing oxidative stress. Future directions may be focused on copper-targeted interventions, such as copper ionophores, with other treatments for more effective anticancer therapies. Co-targeting ferroptosis and cuproptosis by disrupting mitochondria and the labile iron/copper pool can be a potential strategy. Hopefully, the profound applications of ferroptosis and cuproptosis in precision medicine will become evident, which will enable the creation of tailored therapies targeting the unique iron and copper metabolisms of individual tumors. This aligns with the ongoing shift towards personalized oncological therapeutics, contributing to innovative solutions and improved clinical outcomes. However, it is necessary to be more cautious to determine a proper balance between iron and copper metabolism when therapeutic interventions are given to minimize the side effects and relevant toxicities. Managing systemic iron overload and alleviating copper-induced cuproptosis calls for a nuanced approach, optimizing combination therapies, and exploring targeted delivery systems and biomarkers for side effect monitoring. The future work needs to focus more on advancing precision medicine and rendering drug-resistant cancer cells reversible for personalized interventions, aiming for optimal therapeutic outcomes with minimal harm.

## Figures and Tables

**Figure 1 cancers-16-00512-f001:**
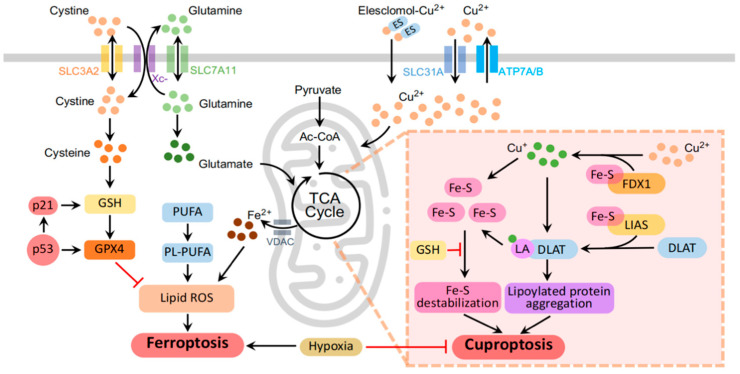
Mechanisms of ferroptosis and cuproptosis. The molecular network framework illustrates mitochondria as central hubs orchestrating the intersection of ferroptosis and cuproptosis. (1) Mitochondria, serving as dynamic pools for iron, undergo iron-induced irreversible changes during ferroptosis (**left** panel). (2) Elevated copper levels induce cuproptosis through FDX1-driven mitochondrial proteotoxic stress in the TCA cycle (**right** panel). (3) Despite entirely different responses, both ferroptosis and cuproptosis, akin to mitochondria, exhibit heightened sensitivity to hypoxia (**middle** panel). The molecular and metal iron transporters are represented by colored channels in the schematic. Colored dots highlight key molecules, emphasizing the dynamic iron and copper labile pools within the cytoplasm. Black arrows delineate stimulatory reactions driving the orchestrated progression of regulated cell death pathways, while red lines indicate inhibitory reactions. The schematic was generated using PowerPoint 2021.

**Table 1 cancers-16-00512-t001:** Clinical trials targeting ferroptosis and iron metabolisms.

Tumor Type	Agent	Role	Phases	Indication	Refs
Breast cancer	Sorafenib	System Xc- inhibitors	phase II	Metastatic BC	NCT00101400
	Artesunate	Iron oxidizing agents	phase I	Metastatic BC	NCT00764036
	Neratinib	Iron activators	phase II	HER2 positive BC	NCT00300781
			phase II	ER positive BC	NCT05933395
	DFO	Iron chelators	phase II	Metastatic Triple Negative BC	NCT05300958
Colon cancer	Mebendazole	System Xc- inhibitors	phase III	Stage IV CRC	NCT03925662
	Sulfasalazine	System Xc- inhibitors	phase III	Metastatic CRC	NCT06134388
	Sorafenib	System Xc- inhibitors	phase II	EGFR positive metastatic CRC with mutant KRAS	NCT00326495
	Curcuminoids	Combined indirect GPX4 inhibitors and HO-1 inducers	NA	Colorectal mucosa of subjects with previously resected adenomatous clonic polyps	NCT00118989
			phase I	CRC	NCT00027495
	Artesunate	Iron oxidizing agents	phase II	Stage II/III CRC	NCT02633098 NCT03093129
Liver cancer	Lcaritin	Others	phase II	Unresectable, non-metastatic HCC	NCT05903456
	Mebendazole/ Lenvatinib	System Xc- inhibitors	NA	Cirrhotics with advanced HCC	NCT04443049
	Ferumoxytol	Iron replacement product	NA	Primary and metastatic liver tumors and hepatic cirrhosis eligible for liver SBRT	NCT04682847
	DFO	Iron chelators	phase I	Unresectable HCC	NCT03652467
Gastric cancer	Sorafenib	System Xc- inhibitors	phase I	Unresectable, recurrent GC	NCT00663741
			phase II	Advanced GC	NCT01187212
Lung cancer	Sorafenib	System Xc- inhibitors	phase II	Relapsed NSCLC	NCT00098254
			phase II	Extensive stage SCLC	NCT00182689
	Cisplatin	GSH inhibitor	Phase IV	Advanced Squamous NSCLC	NCT05312840
			Phase IV	Unresectable advanced, metastatic or recurrent non-squamous NSCLC	NCT02316327
Glioma	L-alanosine	System Xc- inhibitors	phase I/II	High-grade progressive or recurrent malignant gliomas	NCT00075894
	Mebendazole	System Xc- inhibitors	phase I	high-grade glioma	NCT01729260
Prostate cancer	Quercetin/ Genistein	Others	NA	Rising Prostate-specific Antigen	NCT01538316
	Sorafenib	System Xc- inhibitors	phase I	Metastatic, androgen-independent PC	NCT00090545
			phase II	Metastatic or recurrent PC	NCT00093457
	DFO	Iron chelators	phase I/II	STEAP1 positive PCC	NCT01774071
Head and neck cancer	Sorafenib	System Xc- inhibitors	phase II	Recurrent and/or metastatic head and neck cancer	NCT00199160
	5-aminolevulinic acid	Combined indirect GPX4 inhibitors and HO-1 inducers	phase II	Newly diagnosed or recurrent malignancies	NCT05101798
Ovarian cancer	Withaferin A	GPX4 inhibitors	phase I/II	Recurrent OC	NCT05610735
	Neratinib	Iron activators	phase I	Platinum-resistant OC	NCT04502602
Thyroid cancer	Sorafenib	System Xc- inhibitors	phase II	metastatic or recurrent TC	NCT02084732
Melanoma	Buthionine sulfoximine	GCL inhibitor	phase I	Persistent or recurrent stage III malignant melanoma	NCT00661336

**Table 2 cancers-16-00512-t002:** Clinical trials targeting cuproptosis and copper metabolisms.

Tumor Type	Agent	Role	Phases	Indication	Refs
Breast cancer	Disulfiram	Copper supplementation	phase II	Metastatic BC upon failure of conventional systemic and/or locoregional therapies	NCT03323346
			phase II	CTC_EMT positive refractory metastatic hormone receptor positive, HER2 negative BC	NCT04265274
	ATN-224	Copper depletion	phase II	Recurrent or advanced, oestrogen and progesterone receptor positive BC	NCT00674557
	Tetrathiomolybdate	Copper depletion	phase II	BC with moderate-to-high risk of recurrence	NCT00195091
			phase I/II	High risk for relapse triple negative BC	NCT06134375
	Disulfiram	Copper supplementation	phase II	Metastatic BC upon failure of conventional systemic and/or locoregional therapies	NCT03323346
			phase II	CTC_EMT positive refractory metastatic hormone receptor positive, HER2 negative BC	NCT04265274
Prostate cancer	Disulfiram	Copper supplementation	NA	Recurrent PC	NCT01118741
			Phase Ib	Metastatic castrate-resistant PC	NCT02963051
	Elesclomol	Copper supplementation	phase I	Metastatic castration refractory PC	NCT00808418
	ATN-224	Copper depletion	phase II	Relapsed, early-stage PC not on hormone therapy	NCT00405574
			phase II	Hormone refractory PC	NCT00150995
	Tetrathiomolybdate	Copper depletion	phase II	Hormone refractory PC	NCT00150995
Lung cancer	Disulfiram	Copper supplementation	phase II/III	Advanced NSCLC	NCT00312819
	Elesclomol	Copper supplementation	phase I/II	Chemotherapy naive patients with stage IIIB or stage IV NSCLC	NCT00088088
	Tetrathiomolybdate	Copper depletion	phase I	Chemo-naive metastatic or recurrent NSCLC	NCT01837329
			phase I	Stage I-IIIB NSCLC	NCT00560495
Melanoma	Disulfiram	Copper supplementation	phase I/II	Metastatic melanoma	NCT00256230
			Phase Ib	Metastatic melanoma	NCT00571116
	Elesclomol	Copper supplementation	phase III	Metastatic melanoma	NCT00522834
			phase I/II	Metastatic melanoma	NCT00084214
	ATN-224	Copper depletion	phase II	Advanced melanoma	NCT00383851
	Trientine	Copper depletion	phase I	BRAF-mutated metastatic melanoma	NCT02068079
Glioblastoma	Disulfiram	Copper supplementation	phase I/II	Suspected glioblastoma or recurrent glioblastoma undergoing surgical resection	NCT03151772 NCT02715609
			phase II	Recurrent glioblastoma	NCT03034135
			phase II	Newly diagnosed GBM	NCT01777919
			phase II	Unmethylated GBM	NCT03363659
			phase II/III	Recurrent GBM receiving alkylating chemotherapy	NCT02678975
Gastric cancer	Disulfiram	Copper supplementation	NA	Advanced GC	NCT05667415
Pancreatic cancer	Disulfiram	Copper supplementation	phase II	Metastatic PC with rising CA19-9 on Abraxane-Gemzar, FOLFIRINOX, or Gemcitabine	NCT03714555
Liver cancer	Tetrathiomolybdate	Copper depletion	phase II	HCC	NCT00006332
Colon cancer	Tetrathiomolybdate	Copper depletion	phase II	Metastatic CRC	NCT00176774
Ovarian cancer	Trientine	Copper depletion	phase I/II	Latinum-resistant/refractory epithelial ovarian cancer	NCT03480750
Head and neck cancer	Penicillamine	Copper depletion	phase II	Recurrent head and neck cancer	NCT06103617

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
