# Peer review of "Interplay of Ferroptosis and Cuproptosis in Cancer: Dissecting Metal-Driven Mechanisms for Therapeutic Potentials"

_cancers, 2024, doi:10.3390/cancers16030512_

Round 1
Reviewer 1 Report
Comments and Suggestions for Authors
In this review article, the author systematically summarizes and describes the regulatory mechanisms of ferroptosis and cuproptosis, as well as their applications in cancer therapy. The author also summarizes drugs related to ferroptosis and cuproptosis that have entered clinical trials, providing clues for the clinical application of these drugs. However, there are still some aspects that require further in-depth discussion:
-
In comparison to cuproptosis, research on ferroptosis is relatively more extensive. However, in the author's summary of the regulatory mechanisms of ferroptosis, only GPX4 and SLC7A11 are described, and other important proteins involved in the regulation of ferroptosis are not mentioned. A more in-depth and systematic discussion and analysis of these proteins should be conducted.
-
Many drugs do not meet clinical application standards in terms of human toxicity. However, current research suggests that reducing drug concentrations and using them in combination with other drugs can induce cell ferroptosis or cuproptosis. The author should systematically discuss and analyze this aspect when summarizing drugs.
-
The title of the article is "Analysis of the Clinical Applications of Ferroptosis and Cuproptosis," and while the majority of the content discusses the applications of these death mechanisms in cancer treatment, there is no discussion of other relevant diseases.
The Writing is good, I do not have any comments.
Author Response
Thank you very much for taking the time to review this manuscript. Please find the detailed responses below and the corresponding revisions/corrections highlighted/in track changes in the re-submitted files.
Please see the attachment. Many thanks again

Reviewer 2 Report
Comments and Suggestions for Authors
In this review, the authors reported an update on Ferroptosis and Cuproptosis in cancer, highlighting somerelated mechanisms useful in therapeutic strategies. The paper is generally well written and structured, with some interesting reports on clinical trials. However, some major comments to improve the manuscript have been listed below.
1. Some other key enzymes act pivotal roles on ferroptosis and cancer, not discussed by the Authors, i.e. TFRC and Ferritin. Please, improve this section with recent evidences.
2. Figure 1. The figure looks very attractive and impressive, although a little bit "crowded", maybe it could be helpful splitting it into two different figures, on ferroptosis and cuproptosis.
In addition, please, correct "cuproptosis" in the figure and be certain that red lines have the same thickness.
The Figure is not mentioned in the main text.
The extensive form of acronyms in the figure should be listed in the figure legend.
3. Typesetting of the Tables should be improved, it is quite difficult to discern begin/end among the different tumor types.
4. Some acronyms are difficult to understand, maybe not explained in the extensive form at first appearance.
Comments on the Quality of English LanguagePlease, moderate editing of English language should be required throughout the text.
Author Response

(The authors gave the same response as above.)

Reviewer 3 Report
Comments and Suggestions for Authors
This is an interesting review about therapies targeting iron and copper, and their related ferroptosis and cuproptosis. However, a big improvement is expected before it can be considered. Here are a couple of suggestions for further improvement.
- A brief overview of ferroptosis and cuproptosis can be included, such as definition of these two types of cell death. Some confusions exist in the introduction, such as ‘this chapter’ and ‘this analysis’, ‘the chapter’. The introduction is better to be rewritten for this review.
- Some information is repeatedly given which hinders a good reading. For instance, similarity and duplicated information in the introduction and introductory paragraph of Iron/copper in cancers.
- Gaps between sentences are expected to avoided. For example, at the end of ‘2.1 Iron dysregulation in cancer cells’, essential information of drugs have been approved or under development are missing. This kind of gap affects the reading.
- Information provided in 2.1 and 2.2 etc appears to be a brief overview, being summative and lack of details from those specific studies.
- ‘On the other hand,’ can be added before ‘Limiting the labile copper pool…’ in 2.3, line 132.
- The part of ‘3. Mechanisms of ferroptosis and cuproptosis’ can be renamed as ‘Ferroptosis and Cuproptosis’. There are many mechanisms in the subheadings which be omitted. 3.1 Mechanism of Ferroptosis can be 3.1 Ferroptosis. Similarly, this is recommended for the 3.2 Cuproptosis.
- 3.1.1 Iron metabolism, this paragraph appears to be more focusing on targeting iron metabolism to treat cancer rather than iron metabolism itself?
- Certain words are repeatedly used in a paragraph, such ‘venue’ in 3.13. Information/findings are presented as isolated pieces of knowledge rather than a coherent presentation. Adding conjunction words to rewrite this paragraph improve the readability.
- Session 5 CSCs are emphasized but lack of direct evidence to support. Title for this session needs to be reworded to reflect the content.
As seen in the comments
Author Response
Thank you sincerely for dedicating your time to reviewing our manuscript. Enclosed are our detailed responses, along with the corresponding revisions and corrections that have been highlighted or tracked in the resubmitted files.
Please see the attachment. Many thanks again

Round 2
Reviewer 2 Report
Comments and Suggestions for Authors
The Authors provided extensive revision. I suppose the Manuscript is suitable for publication